# Therapeutic Vaccines Targeting Neoantigens to Induce T-Cell Immunity against Cancers

**DOI:** 10.3390/pharmaceutics14040867

**Published:** 2022-04-15

**Authors:** Shih-Cheng Pao, Mu-Tzu Chu, Shuen-Iu Hung

**Affiliations:** 1Cancer Vaccine & Immune Cell Therapy Core Lab, Department of Medical Research, Chang Gung Memorial Hospital, Linkou Branch, No. 5, Fuxing St., Taoyuan 333, Taiwan; bigan818@gmail.com (S.-C.P.); clairechuymu@gmail.com (M.-T.C.); 2Institute of Pharmacology, School of Medicine, National Yang Ming Chiao Tung University, No. 155, Sec. 2, Linong Street, Taipei 112, Taiwan

**Keywords:** cancer therapy, neoantigens, immunotherapy, vaccine

## Abstract

Cancer immunotherapy has achieved multiple clinical benefits and has become an indispensable component of cancer treatment. Targeting tumor-specific antigens, also known as neoantigens, plays a crucial role in cancer immunotherapy. T cells of adaptive immunity that recognize neoantigens, but do not induce unwanted off-target effects, have demonstrated high efficacy and low side effects in cancer immunotherapy. Tumor neoantigens derived from accumulated genetic instability can be characterized using emerging technologies, such as high-throughput sequencing, bioinformatics, predictive algorithms, mass-spectrometry analyses, and immunogenicity validation. Neoepitopes with a higher affinity for major histocompatibility complexes can be identified and further applied to the field of cancer vaccines. Therapeutic vaccines composed of tumor lysates or cells and DNA, mRNA, or peptides of neoantigens have revoked adaptive immunity to kill cancer cells in clinical trials. Broad clinical applicability of these therapeutic cancer vaccines has emerged. In this review, we discuss recent progress in neoantigen identification and applications for cancer vaccines and the results of ongoing trials.

## 1. Introduction

Cancers are driven by genetic instabilities that rapidly accumulate somatic mutations and eventually alter cell properties. Excellent progression has resulted from understanding the mechanisms of genetic mutations, immune recognition of tumor antigens, tumor-mediated immunosuppression, immune surveillance, and tumor escape. Genome sequencing has revealed the heterogeneity of cancer cells, as evidenced by the Cancer Genome Atlas [1,2,3,4]. Neoantigens are a group of tumor-specific antigens (TSAs) arising from genetic variations or retroelements and are considered one of the vital characteristics and derivations of cancers. They have aberrant residues caused by gene alterations that are only expressed on tumor cells and serve as ideal foreign targets for recognition by T cells with high-affinity T-cell receptors (TCRs) [3,5,6]. Theoretically, targeting neoantigens avoids unwanted off-target effects and can precisely guide effector cells to tumor cells. Neoantigen vaccination could be an active immunotherapy and provide immunogens to the immune system to elicit an antitumor immune response. Cancer vaccines have been rapidly developed as a practical method to boost target-specific humoral and cellular immunity and induce long-lasting immune protection [7]. Various vaccination approaches are under investigation and are broadly categorized based on their design methods, including tumor lysates, cell-based vaccines, gene-based vaccines, and peptide-based vaccines. This review summarizes the current knowledge, development, and challenges associated with immunotherapeutics targeting neoantigens by assessing current cancer clinical trials of vaccines to provide insights into the clinical development of personalized cancer immunotherapy.

## 2. Types of Cancer Antigens

Tumor cells have a wide range of protein-expression profiles that differ from those of normal cells. There are different types of tumor antigens: tumor-associated antigens (TAAs), TSAs, and unconventional antigens (UCAs) [8]. 

Compared to normal cells, TAAs are unmodified self-proteins that are abnormally expressed in cancerous cells due to oncogenic signaling processes. The upregulated expression of these wild-type proteins or glycoproteins enables TAAs to act as self-antigens on tumor cells. Most TAAs refer to overexpressed tumor antigens, for example, the receptor for advanced glycation endproducts-1 (RAGE-1), human telomerase reverse transcriptase (hTERT), human epidermal growth factor receptor 2 (HER2), mesothelin, and mucin 1 (MUC1) [9,10,11,12,13,14]. In addition, TAAs can be cell-lineage-differentiation antigens (e.g., prostate-specific antigen (PSA) and prostatic-acid phosphatase (PAP), which are typically not expressed in adult tissues [15,16], and cancer/germline antigens (also known as cancer/testis) (e.g., melanoma-associated antigen 1 (MAGE-A1), melanoma-associated antigen 3 (MAGE-A3), New York esophageal squamous-cell carcinoma 1 (NY-ESO-1), and preferentially expressed antigen of melanoma (PRAME), which are typically only expressed in immune-privileged germline cells [17,18,19,20,21]. These TAAs may represent universal targets for chimeric-antigen-receptor-T (CAR-T) therapy in patients with the same malignancy.

In comparison, TSAs are neoantigens expressed by cancer cells. The uniqueness of the mutant epitopes makes them more likely to be identified by the diverse TCRs of T cells, which are not depleted during clonal selection in the thymus. The degraded peptide fragments of mutant proteins become tumor antigens that play essential roles in T-cell-mediated immunity against cancer. Neoantigens could represent the differences between the peptide repertoires of the major-histocompatibility-complex (MHC) presentations of cancer cells and normal cells. Most TSAs arise from somatic mutations of non-synonymous single-nucleotide variants, frameshifts, infusion or deletion (INDEL) mutations, gene fusion, splice variants, and retroelements [22]. Unlike TAAs, which are self-antigens not recognized by T cells, TSAs are aberrant proteins absent in T-cell clonal selection during thymus education and are, therefore, more likely to escape the central tolerance mechanism. 

Unconventional antigens (UCAs) originate from aberrant transcription, translation, or post-translational modifications in tumor cells. Some UCAs may be completely tumor specific, whereas others may also occur in normal cells.

## 3. Neoantigen-Induced Antitumor Immunity

Regarding the molecular mechanism, neoantigens are proposed to enhance their immunogenicity by modulating immune synapses in several ways: (1) Compared to wild epitopes, neoepitopes strengthen the TCR–MHC-I stability with higher levels of binding affinities and then result in a robust immune response [23]. (2) The absence of neoepitopes in MHC presentation during T-cell selection in thymus education improves TCR recognition [24]. (3) Flanking residues of neoepitopes interfere with and compete with endogenous peptides on the MHC binding groove [25]. 

In the tumor microenvironment (TME), abundant tumor antigens can be secreted via tumor-derived exosomes, which are further enhanced through tumor-cell death caused by immune modulation, radiotherapy, or chemotherapy [26,27,28]. Tumor-infiltrating antigen-presenting cells (APCs) capture tumor antigens and migrate to regional lymph nodes. The epitopes of the captured antigens presented on human-leukocyte-antigen (HLA) molecules of APCs can initiate the activation and differentiation of tumor-specific CD4^+^ and CD8^+^ T cells in the draining lymph node, resulting in the expansion of effector T cells in secondary lymphoid organs [29]. Many effector cells then travel through the bloodstream to the tumor site by involving various chemokines (e.g., C-C motif chemokine ligand 2 (CCL2), C-X-C motif chemokine ligand 2 (CXCL2), and C-X-C motif chemokine ligand 16 (CXCL16)) [30,31]. Activated CD8^+^ T cells can recognize the expressed neoepitope–HLA complexes on tumor cells and then kill cancer cells through the degranulation of cytotoxic proteins, such as perforin, granzyme, and granulysin. CD4^+^ T cells indirectly modulate antitumor cellular and humoral immune responses. Released tumor antigens prime more tumor-reactive immune cells into the TME and trigger adaptive-immune memory responses [29]. However, these immune reactions can be inhibited by an immunosuppressive microenvironment.

## 4. Neoantigen Identification

Approaches for identifying neoantigens and proceeding to vaccine manufacture are illustrated in Figure 1, which involve the discovery of the mutanome by next-generation sequencing (NGS), prediction of HLA epitopes by algorithms or mass spectrometry (MS), and functional validation by immunological assays.

### 4.1. Discovery of the Mutanome by Next-Generation Sequencing

Practitioners and researchers used to have technical restrictions until the advent of advanced high-throughput NGS technologies. Reliable sequencing data are generated at a lower price with greater accuracy to identify individual gene variations in tumor samples. Identifying the entirety of somatic cancer mutations in an individual tumor, referred to as variant calling of the mutanome, yields potential neoantigens. Typically, a small fraction of tumor biopsies is required for DNA and RNA sequencing to obtain the variation profile of the tumor. For SNVs, INDELs, and gene fusions, variants of the mutanome could be detected by comparing WES data from tumor tissue and healthy samples (e.g., PBMC) of the same individual to exclude germline variants [32]. Endogenous retroelement-derived antigens were identified from RNA-expression data. For splicing variants, tools such as SplAdder [33], SpliceGrapher [34], and ASGAL mainly compare the spliced alignments of RNA-seq data to genome references and then generate splicing graphs to predict alternative cleaving events [35]. In addition, these tools integrate proteomic databases to analyze cancer-specific germline and somatic mutations that are rapidly developing. A recently reported proteogenomic tool called QUILTS can be used to generate variants including SNVs, INDELs, fusions, and junctions from RNA-seq data [36].

### 4.2. HLA-Epitope Calling by Computational Algorithms

Only a small portion of expressed neoantigens can fit perfectly into the binding pockets of HLA molecules and possess adequate immunogenicity to elicit immune responses. Selecting neoepitopes with the highest probability of increasing tumor-specific immune responses is critical for designing neoantigen vaccines [6]. The prediction of neoepitopes using computational algorithms is a commonly applied methodology. These programs (e.g., NetMHCpan [37], MULTIPRED [38], IEDB [39], and EpitoolKit [40]) simulate the binding affinity of antigen epitopes with the MHC alleles of subjects and predict potential neoantigens.

The MHC-I-epitope-prediction algorithms have gained greater attention because hypotheses assume that CD8^+^ T-cell-mediated immune responses play a more vital role in antitumor immunity [41]. CD8^+^ T cells infiltrating the tumors has correlated with a better prognosis of the disease [42]. Furthermore, CD4^+^ T cells also play essential roles in cancer immunity. Kreiter et al. reported that mutant MHC-class-II epitopes could drive CD4^+^ T-cell-mediated therapeutic immune responses to cancer [43]. Trans et al. reported that the application of adoptive cell therapy (ACT) using neoantigen ERBB2 (HER2) interaction protein-specific CD4^+^ tumor-infiltrating lymphocytes (TILs) achieved tumor regression in a patient with metastatic cholangiocarcinoma [44]. Due to these clinical findings indicating that CD4^+^ T-cell-mediated antitumor immunity is indispensable for cancer immunotherapy, MHC-II-epitope predictors have been recently improved. For instance, NetMHCIIpan adapted the NN-align algorithms, which add the influence of the core structure of epitopes and the flanking-region characteristics, thereby substantially facilitating MHC-II-binding-prediction performance [45,46].

### 4.3. Identification of HLA Epitopes by Mass Spectrometry (MS)

Recent developments in MS-based sequencing technology have expanded the detection of peptide epitopes on MHC molecules [47,48]. For MS detection, HLA molecules from harvested cell lines or resected tumors can be isolated by pan-HLA immunoprecipitation (IP). After several washes to remove the unwanted mixture, binding epitopes of HLA molecules can be dissociated, purified, and subsequently analyzed by liquid chromatography–tandem MS (LC–MS/MS) [49]. Algorithms have been developed for immunogenic antigen discovery and the establishment of high-resolution, raw quantitative MS data for the patient-customized peptide repertoire, such as MaxQuant [50], SWATH-MS [51], Proteome discovery [52], and PEAKS studio [53]. MS-based sequencing enables researchers to directly identify clinically relevant neoepitopes in human cancer tissues. MS-based HLA-immunopeptidome profiling is also practical for spotting epitopes from post-translational modification [54]. For instance, a study revealed 11 epitopes from gene variants of over 90,000 immunopeptides identified from melanoma patients. Through MS analysis, phosphorylated immune epitopes were identified, and positions 4 and 6 of the 9–12-mer HLA-binding peptide were the major phosphorylation sites [55].

### 4.4. Prediction of HLA Epitopes by Machine Learning Algorithms

By combining it with experimental HLA-immunopeptidome profiling, machine learning in silicon algorithms was developed to provide a rapid and accurate prediction platform. Abelin et al. developed a neural-network prediction algorithm using an extensive dataset collected via MS profiling of HLA-associated peptidomes and found that it outperformed the experimentally measured epitope affinities [56]. GibbsCluster, another machine-learning model built on MS-analysis data integrated with in vitro binding-affinity results, showed an outstanding performance for predicting antigen-restricted epitopes [57,58]. In addition, Bulik-Sullivan et al. launched a computational model named EDGE for epitope prediction, which was trained using a dataset of HLA–MS neoantigen peptides and genomic data of 74 patients. EDGE validation showed a nine-fold-higher positive predictive value than that obtained from tumor test sets using binding-affinity data [59]. The in silico ligand-prediction algorithms ameliorated the previously high false-discovery rate of predicted ligands of specific HLA alleles. Nevertheless, considerable experimental data are required to train the algorithms, especially for the less prevalent HLA alleles for which there are not enough data on epitope affinity or MS results. The sensitivity of algorithms varied among different types of gene alterations and committed bioinformatics tools to optimize HLA-molecule-binding epitope prediction. The Human Immuno-Peptidome Project Consortium aims to establish a repertoire of peptides presented by HLA molecules to facilitate data collection [60]. With more disclosure of epitope sequences, a steadier immunopeptidome database will provide reliable and trustworthy predictions. Table 1 lists the methods and platforms that are commonly used to predict neoantigens.

## 5. Neoantigen-Derived Cancer Vaccines

### 5.1. Tumor Lysates and Allogeneic Tumor-Cell-Based Vaccine

Autologous tumor lysates or allogeneic tumor cells obtained from patients were the earliest developed cancer vaccines. By administering either inactivated resected tumor lysates or allogeneic tumor-cell lysates with additional components such as adjuvants and cytokines, these cancer vaccines could present epitopes of tumor antigens to activate both CD4^+^ and CD8^+^ T cells in the human body [69,70,71].

An autologous tumor-lysate vaccine from Vaccinogen Inc, OncoVax, which uses Bacillus Calmette-Guerin (BCG) as an adjuvant, was shown to extend the recurrence-free period and reduce the risk for recurrences in surgically resected patients with stage II colon cancer. Their phase Ⅲ trial (NTC02448173) evaluating further clinical benefits of OncoVax is ongoing [72]. GVAX (Cell Genesys, Inc., South San Francisco, CA, USA) is an allogeneic whole-tumor-cell vaccine that consists of two prostate-cancer cell lines, LNCaP and PC-3, transfected with a human granulocyte-macrophage-stimulating factor (GM-CSF) gene. Phase I/II studies demonstrated its safety and clinical activity; however, it failed to reach clinical efficacy in a phase III trial of advanced prostate cancer [73]. To improve the overall survival rate, GVAX was recently used with chemotherapy agents and ipilimumab to treat metastatic pancreatic cancer in the trial stage [74]. Other studies on tumor-cell vaccines include melacine (an allogenic melanoma tumor-cell-lysate vaccine) [75], canvaxin (an antigen-rich allogeneic whole-cell vaccine developed from three melanoma cell lines) [76], and TRIMELVax (a heat-shocked melanoma-cell-lysate vaccine) [77]. Although all epitopes are included in this type of vaccine, the contents of neoantigens are quite low, and most are wild-type endogenous peptides, which might dilute the expected immune responses and increase the risk of adverse reactions. Research on optimizing this approach, such as combination therapy and optimized carriers to transport the cells, might address the current limitations of tumor lysates or allogeneic tumor-cell-based vaccines.

### 5.2. DNA-Based Vaccines

DNA vaccines can be introduced into cells and tissues via non-viral or viral gene-delivery systems. After being introduced into the cytoplasm, DNA migrates to the nucleus and initiates the production of antigens. Physical forces mainly represent the non-viral methods of facilitating intracellular gene transfection by transiently loosening the cell-membrane structure. These systems include electroporation, microinjection, and a gene gun to transfect plasmid DNA into the tissue [78]. Although the physical delivery system offers highly efficient gene transfection, tissue damage resulting from the applied physical forces may cause low activity [79]. GNOS-PV02, a neoantigen-DNA vaccine with plasmid-encoded IL-12 administered by electroporation and intradermal injection, entered a phase I/II clinical study with the combination of pembrolizumab for the treatment of advanced hepatocellular carcinoma. The up-to-date result revealed that the objective response rate (ORR) was 25% without reported dose-limiting toxicities (DLTs). Post-vaccination TCR-repertoire analysis identified novel expanded T-cell clones in both peripheral blood and tumor tissue, which potentially mediated the observed regression of tumors [80].

DNA vaccines can also be delivered by viral carriers such as adenoviruses, modified vaccinia viruses, lentiviruses, and retroviruses. The adenovirus is a non-envelope, double-stranded DNA virus commonly used as a viral vector among these viruses. Adenoviral-vector vaccines replace genes that enable replication of transgenes or other genes of interest, making the vector unable to generate their genome copies after delivery. This property also provides the virus with a higher package capacity to incorporate large transgene sequences [81]. Compared to other virus-based vectors, adenoviral vectors have less potential genotoxicity and have been applied to infectious diseases such as COVID-19 [82], Ebola virus [83], and malaria [84]. Nous-209 is a virus-based cancer vaccine encoding 209 commonly shared frameshift mutations of microsatellite instability tumors and uses the Great Ape Adenoviruses vectors for priming and Modified Vaccinia Ankara vectors for boosting. The preliminary results of the phase I study combined with pembrolizumab showed no DLTs. Seven out of the twelve enrolled patients had confirmed partial responses (PRs), and two patients had stable disease (SD), suggesting that Nous-209 is safe and immunogenic and may contribute to early clinical outcomes [85]. PRGN-2009, a human papillomavirus (HPV) therapeutic vaccine encoding 35 non-HLA-restricted epitopes of HPV 16 and 18 by a novel gorilla adenoviral vector, increased the number of T cells targeting HPV 16 or HPV 18 after vaccination in all six recruited patients in a phase I study without observed DLTs [86]. However, pre-existing immunity against particular virus serotypes prevents the efficacy of virus-based vaccines [87]. This problem may be overcome using viral vectors derived from other species [88]. Nonetheless, it remains to be determined whether existing immunity will decrease the immunization potential for a repeated dose of vaccine constructed in the same or similar serotype virus.

In addition to viral vectors, microbes are also candidates for carrying target antigens. Lm-platform technology is an antigen delivery platform via *Listeria monocytogenes* developed by ADVAXIS. Attenuated *Listeria monocytogenes* carrying the bacterial vector expresses fusion proteins containing adjuvant parts and target antigens to T cells after phagocytosis. ADXS-503 is a phase I study of pembrolizumab plus the Lm vaccine targeting 11 common hotspot mutations and 11 TAAs of metastatic non-small-cell lung carcinoma (NSCLC). Antigen-specific T cells were found in all patients with a transient release of pro-inflammatory cytokines. Seven of the nine recruited patients also showed antigen spreading. The ORR was 11%, and the disease-control rate (DCR) was 44%, with one achieving a PR and three achieving SD. The vaccine was well-tolerated without reported immune-related adverse events (irAEs) [89]. Another phase I study, ADXS-NEO-2, targeted personalized neoantigens for each cancer patient. Preliminary findings included immune-cell proliferation, antigen-specific T-cell response, and antigen spreading in one patient at 108 colony-forming units (CFUs). However, two patients had manageable DLTs at an initial dose of 109 CFUs, and the current state of this trial remains unclear [90]. The neoantigen-DNA-vaccine trials currently in the active or completed stages are listed in Table 2.

### 5.3. mRNA-Based Vaccines

Additionally, mRNA vaccines have shown substantial potential against diseases during the COVID-19 pandemic [91]. Theoretically, mRNA vaccines are internalized in the cytoplasm, and antigens of interest can be translated without mutagenesis concerns. The magnitude and rate of mRNA translation are typically higher than those of DNA vaccines. Currently, mRNA can be rapidly produced using in vitro transcription (IVT) methods, making it feasible for scale-up manufacturing. These characteristics make mRNA vaccines powerful tools for responding to emergent needs. 

The significant clinical breakthrough of the application of mRNA cancer vaccines was first published by Sahin et al. [92]. Thirteen patients with stage III and IV melanoma received at least eight doses of personalized neoantigen vaccines percutaneously into the inguinal lymph nodes. Each patient’s five–ten mutations were selected based on the predicted high-affinity binding to autologous HLA class I and HLA class II. Not only were de novo immune responses observed, but pre-existing immune responses against predicted neoantigens were also augmented in all patients. Eight patients remained recurrence-free during the follow-up period. One patient experienced a complete response of metastases, which contributed to neoantigen-vaccine monotherapy. Another patient had a rapid, complete response within two months with PD1-blockade combination therapy. These results translated into sustained progression-free survival (PFS) and significantly reduced the cumulative sum of metastatic events compared to those before vaccine treatment. Notably, immune escape was observed in one patient who initially had a PR but suffered from metastasis two months after 12 vaccinations and follow-up surgeries. Loss of β-2 microglobulin was observed in autologous tumor cells, leading to HLA-class-I dysfunction [92]. 

Additionally, mRNA-4157 is the neoantigen-mRNA-vaccine trial of Moderna and is currently under phase I evaluation for solid tumors. From the updated outcome, the vaccine’s safety was acceptable, with only mild-related adverse events reported [93]. Remarkably, the response rate was 50% for HPV-negative head and neck squamous-cell carcinoma combined with pembrolizumab, and the median PFS was compared favorably to pembrolizumab monotherapy. In addition, 14 of 16 patients with resected solid tumors receiving vaccine monotherapy remained disease free. The trial is ongoing for efficacy analysis [94]. However, the other trial of neoantigen-mRNA vaccines, mRNA-4650, did not proceed because no clinical response was observed. In this study, neoepitopes for each patient were selected by HLA-I prediction and validated by TIL–APC coculture, plus any mutations in the hot driver genes of Kirsten rat sarcoma virus (KRAS), tumor protein p53 (TP53), and phosphatidylinositol-4,5-bisphosphate 3-kinase catalytic subunit alpha (PIK3CA). Despite the suboptimal clinical results, T-cell reactivity against several predicted neoepitopes was found in the post-vaccination PBMC of some patients. TCR analysis revealed neoantigen-specific clonotypes capable of recognizing designed neoantigens, suggesting that a combination of immune-checkpoint inhibitors (ICIs) or immune-cell therapy could have clinical benefits [95]. 

Naked RNA is vulnerable to extracellular RNAse and can undergo rapid degradation that limits the internalization of the vaccine. Improved mRNA-delivery systems facilitate vaccine protection, distribution, and release. For instance, ionizable lipid nanoparticles (LNPs) are self-assembled particles commonly used for RNA delivery. LNPs are stable at physiological pH, but the ionizable coated lipid can interact with the ionic endosomal membrane in an acidic endosomal microenvironment, thus promoting membrane fusion and RNA release. Moreover, mRNA has intrinsic immunogenicity, recognized mostly by toll-like receptor-7 and -8, and activates downstream interferon pathways and pro-inflammatory cytokine release. Although this might augment adaptive-immune responses, it could also dampen the antigen presentation. Unwanted double-stranded RNA (dsRNA) produced during IVT can activate RNA-dependent protein kinase, phosphorylate eukaryotic elongation factor-2, and block mRNA translation [96]. Several strategies have been investigated to overcome this limitation. Baiersdörfer et al. presented a dsRNA-removal method using cellulose in an ethanol-containing buffer. Up to 90% of dsRNA contaminants can be removed, resulting in better translation efficacy in vivo [97]. CureVax AG developed an RNA/protamine complex that serves as a toll-like receptor 7/8 (TLR7/8) adjuvant, increasing antitumor immunity after vaccination [98]. Luo et al. reported a formulation of synthetic polymeric nanoparticles with an intrinsic activating property for the stimulator of interferon genes (STING), leading to the inhibition of tumor progression in three types of cancer models [99]. In addition, BioNTech developed an RNA-lipoplex cancer-vaccine platform, Lipo-MERIX, which can precisely target dendritic cells (DC) in the lymphoid compartment by systematic administration (intravenous injection) to induce a potent immune response [100]. Several trials evaluating Lipo-MERIX carrying TAA or TSA for different types of solid tumors are ongoing. A relative trial targeting TAA for advanced melanoma, BNT-111, has recently received FDA fast-track designation [101]. Active and completed neoantigen-mRNA-vaccine trials are listed in Table 3.

### 5.4. Protein and Peptide Vaccines

Peptide-based vaccines use synthetic peptides to trigger peptide-specific immune responses against cancer. It is intuitive and cost-effective, and no intricate logistics are required for transport and restoration. As reviewed by Shemesh et al., neoantigen vaccines derived from peptides, along with mRNA, have undergone the most ongoing clinical trials [102]. The primary outcomes of peptide vaccines showed promising results in treating melanoma and brain malignancies in multiple trials [103,104]. 

Hilf et al. conducted the GAPVAC trial for glioblastoma by administering peptide vaccines containing the predicted neoantigens and glioma-related TAAs. Notably, Th1 cells were induced in 11 of 13 patients receiving the neoepitope vaccine. In one patient who had a complete response after vaccination but experienced recurrence two years afterward, high infiltration by T cells was found, with a favorable ratio of CD8^+^/FOXP3^+^ (forkhead box P3+) Treg cells from the re-resected tumor [105]. Similar results were reported by Keskin et al., who demonstrated that neoantigen-specific CD4^+^ and CD8^+^ T cells enriched in the memory phenotype were found after neoantigen-peptide administration. This study further proved that neoantigen-specific T cells triggered by the vaccine could migrate into intracranial glioblastoma tumors [103]. 

Recently, Platten et al. tested the safety and efficacy of a mutated isocitrate dehydrogenase 1 (IDH1) peptide vaccine in a phase I trial. Mutations in IDH1 are molecular characteristics of certain gliomas that contribute to the early stages of tumor development. Patients with the IDH1 R132H variant were recruited and treated with a 20-mer peptide containing a mutated spot. A mutant-specific T-cell response was found in over 90% of recruited patients with appropriate safety profiles [106]. In recent years, elongated CD8^+^ T-cell epitopes have been thought to enhance epitope-specific anticancer immunity. Unlike the predicted short epitopes, long peptides are believed to only be processed and presented by professional APC, leading to robust T-cell induction. In the mutant IDH1 trial, a single LSP (long synthetic peptide) was presented across various MHC alleles and, therefore, could be applied as an off-the-shelf product. 

Moreover, the combination of neoantigen-peptide vaccines and ICIs has been validated in several trials. The NEO-PV-01 phase Ib clinical trial of a personalized peptide vaccine plus anti-PD1 (anti-programmed death-1) agent was evaluated for safety and efficacy in patients with advanced melanoma, NSCLC, and bladder cancer. Persistent cytotoxic T-cell responses were identified post-vaccination, without severe adverse reactions, in all three cancer cohorts. The median ORR and PFS were favorably compared with historical results for anti-PD-1 monotherapy but could not firmly attribute these outcomes to the vaccine because it was a single-arm investigation [107]. The comparison of neoantigen-peptide-vaccine monotherapy or in combination with ICIs was validated in an ongoing trial, GEN-009 [108]. 

NeoVax is a personalized long-peptide vaccine plus poly-ICLC (polyinosinic-polycytidylic acid stabilized with polylysine and carboxymethylcellulose) (i.e., a TLR-3 and MDA5 (melanoma differentiation-associated protein 5) agonist) [104,107]. A long-term follow-up study revealed that all patients with resected metastatic melanoma who had previous NeoVax treatment were still alive up to four years after treatment. Six of the eight patients had no evidence of disease. T cells with reactivity against certain vaccinated neoantigens persisted in the circulating blood of patients during the priming, boosting, and post-vaccination stages (up to 4.5 years). After the vaccination period, these functional T cells shifted to the less exhausted memory phenotype. Encouragingly, T cells able to target non-vaccinated TAAs or neoantigens were identified only in the post-vaccination sample, suggesting that the neoantigen-peptide vaccine could induce epitope spreading [108]. Epitope spreading has also been observed in several neoantigen-peptide-vaccine trials, including the NEO-PV-01, GEN-009, and glioblastoma trials [103,105,107,108]. In the NeoVax follow-up study, enhanced epitope spreading was observed in one patient experiencing recurrence in the post-vaccination period, but no evidence of disease after pembrolizumab therapy was shown, indicating that the combination of the neoantigen vaccine and ICIs could further improve clinical outcomes [108]. More neoantigen-peptide-vaccine trials in the active and completed stages are summarized in Table 4.

### 5.5. DC-Based Vaccines

The cell-based-vaccine approach exploits autologous DCs loaded with tumor antigens in various formats, including tumor lysates, DNA, mRNA, or peptides. Encouraging results, including Sipuleucel-T, an autologous DC vaccine targeting prostatic-acid phosphatase (PAP), a TAA, have demonstrated a significant improvement in overall survival for men with metastatic castration-resistant prostate cancer and was approved by the FDA [109]. For the neoantigen-pulsed DC vaccine, Carreno et al. conducted a trial applying an in vitro matured autologous DC vaccine stimulated by personalized neoantigen peptides in three patients with advanced melanoma. TCR-sequencing results indicated diverse neoantigen-specific clonotypes induced by personalized DC vaccines, and increased immunity was observed in all patients [110]. Moreover, a patient with metastatic pancreatic cancer experienced regression of multiple metastatic lesions 2.5 months after DC-based-vaccine treatment. In this case, the selected neoepitope was an HLA-A*0201–restricted KRAS-G12D epitope, and the patient received a vaccine containing a neoantigen plus DC and neoantigen-reactive CD8^+^CD137^+^ T cells [111]. Similar research on patients with heavily treated lung cancer by administering a neoantigen-peptide-loaded DC vaccine demonstrated a 25% ORR and 75% DCR. Although none of the recruited patients achieved CR, the results were auspicious considering the initially poor prognosis of the study population. In addition, they noticed that the neoantigen-loaded DC vaccine could re-induce objective responses to ICIs in patients who had a relapse after previous ICI treatment. This finding corresponds to that mentioned in the peptide-vaccine section, namely that the combination of cancer vaccines and ICMs could further provide synergetic therapeutic benefits [112].

## 6. Opinions and Future Perspectives

Therapeutic cancer vaccines have several promising clinical outcomes. However, all vaccines are still in the early stages of clinical trials (phases I and II). This may reflect difficulties in inducing a robust immune response to kill aggressive cancer cells in immunosuppressed patients. In addition, the variation in neoantigens in different individuals makes large-scale applications more challenging than targeting commonly shared antigens. Whether therapeutic vaccines can be applied and used in clinical practice depends on different factors, such as (1) the ability to yield sufficient numbers of T cells to overcome the suppressive TME, (2) augmented immune cells that can penetrate and infiltrate the tumors, (3) the use of adequate adjuvants and carriers, and (4) optimal selection of target antigens [113,114]. 

Moreover, T-cell exhaustion has been reported in numerous studies where vaccine-elicited T lymphocytes often express several inhibitory receptors [92,103,104,105]. A combination of ICIs or other immunotherapies is necessary to achieve synergistic efficacy. In addition to cytotoxic T cells, the importance of CD4^+^ T cells in cancer immunity has been well established. Notably, MHC-II-restricted tumor epitopes also play a crucial role in immunotherapy efficacy. Activated CD4^+^ cells could give rise to the induction of CD8+ T cells with less inhibitory profiles and strengthened effector functions. At the beginning of cancer-vaccine treatment, priming of the immunization determines the phenotype and magnitude of the vaccine-elicited immune response. Ideally, a subset of neoantigen-specific T cells with memory phenotypes is generated after antigen clearance. Continuous exposure to antigens can induce functional profiles of T cells, including memory T cells [115,116,117]. The expression of MHC-II epitopes by tumors can recruit more intratumoral T cells and inducible nitric-oxide-synthase-positive macrophages [118]. Including MHC-II epitopes and stimulants to activate CD4^+^ cells in cancer vaccines has been suggested to improve efficacy. Therefore, optimized priming and boosting regimens for vaccination should be carefully determined. Applying advanced technologies to identify TSAs and generate vaccines with potent adjuvants is the key to developing successful anticancer therapeutics. 

Immunoengineering, the field that integrates nanotechnology, bioengineering, material sciences, drug delivery, and immunology, aims to elicit a robust antitumor immune response. In particular, nanoparticles provide better delivery efficiency and T-cell priming for gene-based and peptide-based vaccines. By loading or conjugating adjuvants, innate-immunity agonists, and target receptors to nanoparticles, co-delivery can enhance the magnitude of antitumor responses [119,120]. For instance, a "nanodiscs" mixing synthetic high-density lipoprotein, cysteine-modified antigens, and cholesterol-modified CpG adjuvant successfully promoted antigen presentation and eliminated established mouse tumors when combined with ICIs [121]. In addition, a biodegradable matrix loaded with small molecules and biologics implanted near the tumor or post-resection sites can reverse the immunosuppressive conditions. The matrix provides artificial immune niches that enable the in situ manipulation of cells [122]. Implantation of a biopolymer-based scaffold loaded with tumor-reactive T cells and agonists enhances antigen presentation and T-cell response to eradicate inoperable orthotropic tumors in mice [123,124]. Moreover, protein-based gels loaded with nanoparticles containing anti-CD47, an inhibitory ligand on cancerous cells, polarized macrophages to M1 phenotypes, and prolonged survival in mice with incomplete resection [125]. Further exploration using matrix-coated tumor neoantigens as cancer vaccines is required. These advanced methods aim to provide the best formulation and dosage of tumor antigens and adjuvants to induce the immune cells and improve the efficacy of therapeutic cancer vaccines.

The immune system is intricate and highly coordinated; the absence of specific cytokines or subsets of immune cells could substantially alter the subsequent cascade of responses, indicating that ex vivo immunostimulatory experiments may not precisely interpret the real circumstances in vivo. Emerging tools such as the three-dimensional modeling system and immune organ/tumor "on a chip" system could foster sophisticated examination of immune-organ function and immune-cell interaction [126]. For example, a microfluidic chip containing hepatocellular carcinoma cells was built to evaluate the time-dependent migration and cytotoxicity of TCR-engineered T cells. The device allowed the investigation of T-cell ability under different inflammatory conditions [127]. In addition, the microphysiological 3D cancer model used to test the efficacy of receptor-engineered cells was validated in lung-, breast-, and ovarian-cancer models [128,129].

Regarding the different types of formulations, mRNA vaccines have the advantage of a cost-effective and straightforward manufacturing procedure. On the other hand, favorable clinical outcomes were also observed in patients who received protein and peptide vaccines, such as NeoVax, Neo-PV-01, GAPVAC, and the IDH1 peptide vaccine for glioma. Targeting neoantigens through integrating immunotherapeutics, including vaccines, cell-based therapy, ICIs, and immunoengineering may provide opportunities to overcome the unmet needs of cancer immunotherapy.

## 7. Conclusions

The development of therapeutic cancer vaccines is a promising prospect for improving the safety and efficacy of the currently used immunotherapeutics. This is a ready-to-produce procedure with an extensive selection of formats. Targeting neoantigens and other TSAs enables immunogens to induce tumor-specific adaptive-immune responses. High-throughput sequencing, epitope-identified mass spectrometry, and predictive algorithms have enabled neoantigen epitopes to be disclosed and subsequently used to design vaccines. Two primary tactics for neoantigen vaccines are evolving. One harnesses personalized vaccines for personalized therapy, and the other utilizes shared neoantigens or viral oncoproteins as off-the-shelf therapeutics. The clinical results summarized in this review indicate encouraging progress in disease control and favorable immune responses. However, several hurdles remain, including on-target distribution, conversion of immunosuppressive environments, and antigen selection. By investigating adequate delivery systems, carriers, adjuvants, and new immunology research tools, these endeavors could gradually reach new heights. Numerous studies using various formats, therapeutic regimens, delivery systems, and combination therapies are still in progress. Targeting neoantigens could be a path to success for significant clinical improvement in cancer treatment.

## Figures and Tables

**Figure 1 pharmaceutics-14-00867-f001:**
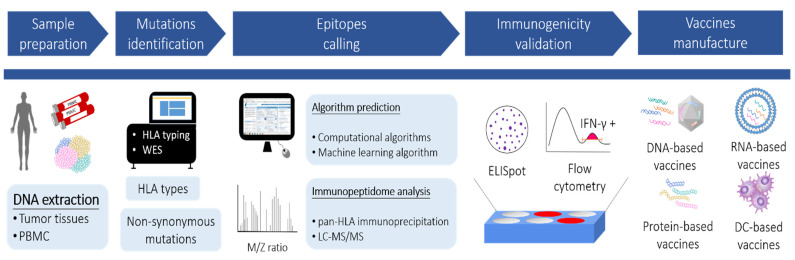
Schematic representation of neoantigen selection for therapeutic cancer vaccines. The DNA samples are extracted from cancer tissues and peripheral blood mononuclear cells (PBMC), respectively. Non-synonymous mutations and HLA types are obtained through whole-exome sequencing (WES) and HLA genotyping following bioinformatic analyses. The potential neoepitopes derived from the identified mutations are prioritized according to (1) algorithm prediction or (2) immunopeptidome analysis. Afterward, each prioritized neoepitope is examined by immunological assays (e.g., ELISpot or flow cytometry) to validate the immunogenicity. Vaccines encoding the selected neoepitopes are then generated in various formats, including DNA-based, RNA-based, peptide-based, and dendritic-cell (DC)-based vaccines.

**Table 1 pharmaceutics-14-00867-t001:** Methods and platforms commonly used for predicting neoantigens.

Method [Ref]	Principle	Year
NetMHCpan [37]	Comparison of epitope sequences by artificial neural networks that provide peptide–MHC-I-affinity predictions	2016
NetMHCIIpan [61]	Pan-specific predictor able to predict binding affinities for all HLA-class-II molecules based on neural networks	2013
MHCflurry [62]	Neutral networks including mass-spectrometry datasets for predicting peptide–MHC-I affinities	2018
ConvMHC [63]	peptide–MHC interactions encoded into image-like array data and analyzed by deep convolutional neural network	2017
PLAtEAU [64]	Defines shared consensus epitopes arising from a series of eluted nested peptides and quantified by mass spectrometry	2018
MuPeXI [65]	Integration of somatic mutation calls, list of HLA types, an optional gene-expression profile, and NetMHCpan 3.0 to provide immunogenicity score based on similarity to non-mutated wild-type peptide	2017
NeoPrepPipe [66]	Predicts neoantigen burdens and provide insights into the tumor heterogeneity, somatic mutation calls, and patient HLA haplotypes	2019
EpitopeHunter [67]	Integrates expression of RNA with artificial neutral networks of immunogenicity-prediction algorithm based on the hydrophobicity of the TCR contact residues	2015
Neopepsee [68]	Integrates sequence and amino-acid-immunogenicity information, including antigen processing and presentation to reduce the false-discovery rate	2018

**Table 2 pharmaceutics-14-00867-t002:** Clinical trials of neoantigen-DNA vaccines.

Trial No. (Brand Name)	Target	Indication	Format/Route of Administration	Combination Therapy	Status
NCT03122106	Personalized NeoAg + Mesothelin	Pancreatic Cancer	Plasmid DNA/Electroporation + IM injection	N/A	Phase 1, Active, Not Recruiting
NCT04015700 (GNOS-PV01)	Personalized NeoAg	Unmethylated Glioblastoma	Plasmid DNA/Electroporation + IM injection	Pembrolizumab, Plasmid encoded IL-12 (INO-9012)	Phase 1, Recruiting
NCT04251117 (GNOS-PV02)	Personalized NeoAg + Mesothelin	HCC	Plasmid DNA/Electroporation + IM injection	Pembrolizumab, Plasmid encoded IL-12 (INO-9012)	Phase 1/2a, Recruiting
NCT04990479 (Nous-PEV)	Personalized NeoAg	Melanoma, NSCLC	Adenovirus vector + Vaccinia virus vector/IM injection	Pembrolizumab	Phase 1, Recruiting
NCT04041310 (Nous-209)	Personalized NeoAg	MSI-H CRC, gastric, G-E junction tumors	Adenovirus vector + vaccinia virus vector/IM injection	Pembrolizumab	Phase 1/2, Active, Not Recruiting
NCT05018273 (VB10.NEO)	Personalized NeoAg	Solid Tumors	Plasmid DNA/IM injection	Atezolizumab	Phase 1b, Recruiting
NCT02348320	Personalized NeoAg	Triple-Negative Breast Cancer	Plasmid DNA/Electroporation + IM injection	N/A	Phase 1, Completed
NCT03953235 (SLATE)	Shared Neoantigen	Shared neoantigen positive tumors	Adenovirus vector + RNA vector/Not specific	Nivolumab, Ipilimumab	Phase 1/2, Recruiting
NCT03265080 (ADXS-NEO)	Personalized NeoAg	Colon Cancer, Head & Neck Cancer, NSCLC, Urothelial Carcinoma, Melanoma	*Lm*-based vector/I.V. infusion	Pembrolizumab (selectively)	Phase 1, Active, Not Recruiting
NCT03847519 (ADXS-503)	Personalized NeoAg	NSCLC, Metastatic SCC, Metastatic NSCLC	*Lm*-based vector/I.V. infusion	Pembrolizumab (selectively)	Phase 1/2, Recruiting

Abbreviations: CRC, colorectal cancer. HCC, hepatocellular carcinoma. I.V., intravascular infusion. I.M., intramuscular injection. *Lm*, *Listeria monocytogenes*. MSI-H, high microsatellite instability. NSCC, small-cell lung cancer. NSCLC, non-small-cell lung cancer.

**Table 3 pharmaceutics-14-00867-t003:** Clinical trials of neoantigen RNA vaccines.

Trial No. (Brand Name)	Target	Indication	Format/Route of Administration	Combination Therapy	Status
RO7198457					
NCT03289962	Personalized NeoAg	Solid tumors	RNA-Lipoplex/I.V.	Atezolizumab	Phase 1a/1b, Recruiting
NCT03815058	Personalized NeoAg	Advanced Melanoma	RNA-Lipoplex/I.V.	Pembrolizumab	Phase 2, Recruiting
NCT04486378	Personalized NeoAg	Colorectal Cancer Stage II, III	RNA-Lipoplex/I.V.	N/A	Phase 2, Recruiting
NCT04161755	Personalized NeoAg	Pancreatic Cancer	RNA-Lipoplex/I.V.	Atezolizumab, mFOLFIRINOX	Phase 1, Recruiting
IVAC mutanome					
NCT02035956	Personalized NeoAg	Melanoma	Not specific/Intra-nodal	RBL001/RBL002 (TAA RNA Vaccine)	Phase 1, Completed
NCT02316457	Personalized NeoAg	Breast Cancer (TNBC)	Nanoparticulate lipoplex RNA/I.V.	IVAC_W_bre1_uID (TAA RNA vaccine)	Phase 1, Active, Not Recruiting
mRNA-4157					
NCT03897881	Personalized NeoAg	Melanoma	lipid encapsulated RNA/I.M.	Pembrolizumab	Phase 2, Active, Not Recruiting
NCT03313778	Personalized NeoAg	Solid tumors	lipid encapsulated RNA/I.M.	Pembrolizumab	Phase 1, Recruiting
mRNA-5671					
NCT03948763	KRAS common mutations	Solid Tumors	lipid encapsulated RNA/I.M.	Pembrolizumab (selectively)	Phase 1, Recruiting

Abbreviations: I.V., intravascular infusion. I.M., intramuscular injection. TAA, tumor-associated antigens. TNBC, triple-negative breast cancer.

**Table 4 pharmaceutics-14-00867-t004:** Clinical trials of neoantigen-peptide vaccines.

**Trial No. (Brand Name)**	**Target**	**Indication**	**Format/Route of Administration**	**Combination Therapy**	**Status**
NCT04799431	Personalized NeoAg	MMR-p Colon Cancer Pancreatic Ductal Cancer	Peptide + poly-ICLC/subcutaneous	Retifanlimab	Phase 1, Not Yet Recruiting
NCT03956056	Personalized NeoAg + Mesothelin	Pancreatic Cancer	Peptide + poly-ICLC/ subcutaneous	N/A	Phase 1, Recruiting
NCT04248569	DNAJB1- PRKACA fusion	Fibrolamellar Hepatocellular Carcinoma	Peptide + poly-ICLC	Nivolumab, Ipilimumab	Phase 1, Recruiting
NCT04117087	Common mutant KRAS	Colorectal Cancer Pancreatic Cancer	Peptide + poly-ICLC	Nivolumab, Ipilimumab	Phase 1, Recruiting
NCT04749641	Histone H3.3-K27M mutant	Diffuse Intrinsic Pontine Glioma	Peptide + poly-ICLC/subcutaneous	N/A	Phase 1, Recruiting
NCT03715985 (NeoPepVac)	Personalized NeoAg	Melanoma, NSCLC, Bladder, Urothelial Carcinoma,	Peptide + CAF09b/I.P. + I.M.	N/A	Phase 1, Recruiting
NCT03359239 (PGV-001)	Personalized NeoAg	Urothelial/Bladder Cancer	Peptide + poly-ICLC	Atezolizumab	Phase 1, Recruiting
NCT02149225 (GAPVAC)	Personalized NeoAg	Glioblastoma	Peptide + poly-ICLC/not specific	TAA peptide vaccine, GM-CSF	Phase 1, Completed
NeoVax					
NCT01970358	Personalized NeoAg	Melanoma	Peptide + poly-ICLC/subcutaneous	N/A	Phase 1, Completed
NCT02950766	Personalized NeoAg	Kidney cancer	Peptide + poly-ICLC/subcutaneous	Nivolumab, Ipilimumab	Phase 1, Recruiting
NCT02287428	Personalized NeoAg	Glioblastoma	Peptide + poly-ICLC	Pembrolizumab Temozolomide (Both selectively)	Phase 1, Recruiting
NCT03929029	Personalized NeoAg	Melanoma	Peptide + poly-ICLC + Montanide	Nivolumab Ipilimumab	Phase 1b, Recruiting
NCT0402487	Personalized NeoAg	Ovarian Cancer	Peptide + poly-ICLC	Nivolumab	Phase 1, Recruiting
NCT03219450	Personalized NeoAg	Lymphocytic Leukemia	Peptide + poly-ICLC	Pembrolizumab Cyclophosphamide (both selectively)	Phase 1, Recruiting
Neo-PV-01					
NCT03380871	Personalized NeoAg	Lung cancer	Peptide + poly-ICLC/subcutaneous	Pembrolizumab Carboplatin Pemetrexed	Phase 1, Completed
NCT02897765	Personalized NeoAg	Urinary Bladder Cancer Melanoma Lung Cancer	Peptide + poly-ICLC/subcutaneous	Nivolumab	Phase 1, Completed

Abbreviations: I.V., intravascular infusion. I.M., intramuscular injection. MMR-p, mismatch repair protein deficiency. NSCLC, non-small-cell lung cancer. poly-ICLC, polyinosinic-polycytidylic acid. TAA, tumor-associated antigens.

## Data Availability

Not applicable.

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
