# Peer review of "Therapeutic Vaccines Targeting Neoantigens to Induce T-Cell Immunity against Cancers"

_pharmaceutics, 2022, doi:10.3390/pharmaceutics14040867_

Round 1

Reviewer 1 Report

The authors wrote a review of the current development of anti-cancer vaccines and provided the researchers a summary of important information. However, it will be better for the authors to discuss why all the vaccines are still at the early stage (Phase I and II)? And what will be the solution to it? What kind of vaccine will be the most promising for cancer treatment?

Author Response

Reply: We have added this discussion in the section 6 (6. Opinions and future perspectives).

  1. Opinions and future perspectives

Therapeutic cancer vaccines have several promising clinical outcomes. However, all vaccines are still in the early stages of clinical trials (phases I and II). This may reflect difficulties in inducing a robust immune response to kill aggressive cancer cells in immunosuppressed patients. In addition, the variation in neoantigens in different individuals makes large-scale applications more challenging than targeting commonly shared antigens. Whether therapeutic vaccines can be applied and used in clinical practice depends on different factors, such as (1) the ability to yield sufficient numbers of T cells to overcome the suppressive TME, (2) augmented immune cells that can penetrate and infiltrate into the tumors, (3) the use of adequate adjuvants and carriers, and (4) optimal selection of target antigens [113, 114].

Moreover, T cell exhaustion has been reported in numerous studies where vaccine-elicited T lymphocytes often express several inhibitory receptors [92, 103-105]. A combination of ICIs or other immunotherapies is necessary to achieve synergistic efficacy. In addition to cytotoxic T cells, the importance of CD4+ T cells in cancer immunity has been well established. Notably, MHC II-restricted tumor epitopes also play a crucial role in immunotherapy efficacy. Activated CD4+ cells could give rise to the induction of CD8+ T cells with less inhibitory profiles and strengthened effector functions. At the beginning of cancer vaccine treatment, priming immunization determines the phenotype and magnitude of the vaccine-elicited immune response. Ideally, a subset of neoantigen-specific T cells with memory phenotypes is generated after antigen clearance. Continuous exposure to antigens can induce functional profiles of T cells, including memory T cells [115-117]. The expression of MHC II epitopes by tumors can recruit more intratumoral T cells and inducible nitric oxide synthase-positive macrophages [118]. Including MHC II epitopes and stimulants to activate CD4+ cells in cancer vaccines has been suggested to improve efficacy. Therefore, optimized priming and boosting regimens for vaccination should be carefully determined. Applying advanced technologies to identify TSAs and generate vaccines with potent adjuvants is the key to developing successful anticancer therapeutics.

Immunoengineering, the field that integrates nanotechnology, bioengineering, material sciences, drug delivery, and immunology, aims to elicit a robust antitumor immune response. In particular, nanoparticles provide better delivery efficiency and T cell priming for gene-based and peptide-based vaccines. By loading or conjugating adjuvants, innate immunity agonists, and target receptors to nanoparticles, co-delivery can enhance the magnitude of antitumor responses [119, 120]. For instance, a "nanodiscs" mixing synthetic high-density lipoprotein, cysteine-modified antigens, and cholesterol-modified CpG adjuvant successfully promoted antigen presentation and eliminated established mouse tumors when combined with ICIs [121]. In addition, the biodegradable matrix loaded with small molecules and biologics implanted near the tumor or post-resection sites can reverse the immunosuppressive conditions. The matrix provides artificial immune niches that enable in situ manipulation of cells [122]. Implantation of a biopolymer-based scaffold loaded with tumor-reactive T cells and agonists enhances antigen presentation and T cell response to eradicate inoperable orthotropic tumors in mice [123, 124]. Moreover, protein-based gels loaded with nanoparticles containing anti-CD47, an inhibitory ligand on cancerous cells, polarized macrophages to M1 phenotypes, and prolonged survival in mice with incomplete resection [125]. Further exploration using matrix-coated tumor neoantigens as cancer vaccines is required. These advanced methods aim to provide the best formulation and dosage of tumor antigens and adjuvants to induce the immune cells and improve the efficacy of therapeutic cancer vaccines.

The immune system is intricate and highly coordinated; the absence of specific cytokines or subsets of immune cells could substantially alter the subsequent cascade of responses, indicating that ex vivo immunostimulatory experiments may not precisely interpret the real circumstances in vivo. Emerging tools such as the three-dimensional modeling system and immune organ/tumor "on a chip" system could foster sophisticated examination of immune organ function and immune cell interaction [126]. For example, a microfluidic chip containing hepatocellular carcinoma cells was built to evaluate the time-dependent migration and cytotoxicity of TCR-engineered T cells. The device allowed the investigation of T cell ability under different inflammatory conditions [127]. In addition, the microphysiological 3D cancer model used to test the efficacy of receptor-engineered cells was validated in lung, breast, and ovarian cancer models [128, 129].

Regarding the different types of formulations, mRNA vaccines have the advantage of a cost-effective and straightforward manufacturing procedure. On the other hand, favorable clinical outcomes were also observed in patients who received protein and peptide vaccines, such as NeoVax, Neo-PV-01, GAPVAC, and the IDH1 peptide vaccine for glioma. Targeting neoantigens through integrating immunotherapeutics, including vaccines, cell-based therapy, ICIs, and immunoengineering, may provide opportunities to overcome the unmet needs of cancer immunotherapy.

Reviewer 2 Report

In the review entitled "Therapeutic vaccines targeting neoantigens to induce T cell immunity against cancers" by Pao et.al, the authors collected, present and discuss the current knowledge regarding the ongoing progress of neoantigen identification methodology as well as its application on cancer vaccines development. Additionally they discuss the results of recent and ongoing trials. In general, it is a well organised and prepared work, easily for the reader to follow. It covers a good part of the literature and the important issues of the topic. A language editing is suggested.

Author Response

Reply: Our manuscript has been edited by Elsevier language editing services. A certificate is attached.

Reviewer 3 Report

In general terms this review is well written but authors should add their own personal considerations, therefore, minor revisions are required.

Abbreviations should be defined in parentheses the first time they appear in the abstract and in the main text used consistently thereafter;

Page 2, line 62; such as…

Page 10, line 374: please, write: Notably, Th1 cells;

Page 11, line 405: please, add references;

Author Response

Reviewer 3:

In general terms this review is well written but authors should add their own personal considerations, therefore, minor revisions are required.

Abbreviations should be defined in parentheses the first time they appear in the abstract and in the main text used consistently thereafter;

Page 2, line 62; such as…

Reply: We added the abbreviations.

Page 10, line 374: please, write: Notably, Th1 cells;

Reply: The sentence has been revised.

Page 11, line 405: please, add references;

Reply: We have added the references [104, 107].
